# TRAINING-FREE NATIVE SPARSE ATTENTION FOR KV CACHE COMPRESSION

## ABSTRACT

Large language models (LLMs) suffer from inference inefficiency as KV cache memory and computation scale linearly with context length. Existing KV cache compression methods typically use attention-score-based token-level selection, which leads to uneven attention distributions—overemphasizing prompt boundaries and neglecting global context. We propose the HBW-KV method for training-free KV cache compression with two innovations: (1) block-wise selection that achieves superior precision over token-level approaches, and (2) a hierarchical selection strategy that preserves global context without extra training. Our approach adapts insights from Native Sparse Attention to the KV cache compression setting, enabling plug-and-play integration into existing pre-trained models. Extensive experiments demonstrate significant improvements: 16× compression ratio on 32K sequences, reduces KV cache by over 90%, accelerates decoding by 4×, and maintains over 99%+ accuracy. Our training-free solution offers universal compatibility with existing LLM frameworks for practical long-context applications.

## 1 INTRODUCTION

Recent advancements in large language models (LLMs), such as GPT-4 Achiam et al. (2023) with 128K, Claude-3 Anthropic (2024) with 200K, and Gemini-Pro-1.5 Reid et al. (2024) with 1M, have significantly extended the context length—from tens of thousands to even millions of tokens. Despite these impressive capabilities, processing long-context inputs remains challenging, particularly due to inefficiencies associated with the Key-Value (KV) cache in attention mechanisms Li et al. (2024a). Specifically, during inference, the decoding stage is memory-bounded Shazeer (2019) due to attention computations involving all previously cached KV pairs, causing decoding latency to increase linearly with prompt length. Moreover, the large KV cache consumes substantial memory resources, imposing considerable hardware demands and limiting the scalability of LLMs.

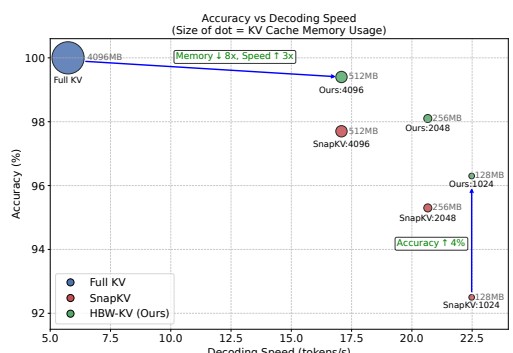

Figure 1: Accuracy and efficiency comparison of different methods. The horizontal axis represents decoding speed, while the vertical axis indicates the relative accuracy compared to the uncompressed baseline.

To address these challenges, previous methods such as StreamingLLM Xiao et al. (2023), H2O Zhang et al. (2023), FastGen Ge et al. (2023), and Scissorhands Liu et al. (2024) have proposed various KV cache optimization strategies, primarily focusing on selective eviction during generation. However, these approaches often overlook the critical issue of compressing KV caches for prompt tokens, which typically represent the primary bottleneck in memory efficiency. Recent studies, notably SnapKV series Li et al. (2024b); Cai et al. (2024); Feng et al. (2024), attempted to tackle this issue by identifying stable attention patterns within prompt tokens at the prefill stage. SnapKV demonstrates the feasibility of predicting critical attention patterns prior to generation, significantly enhancing decoding speed and memory efficiency.

Another promising direction involves designing sparse attention structures, such as Grouped Query Attention (GQA) Ainslie et al. (2023) and Multi-Query Attention (MQA) Shazeer (2019), though these typically require extensive pre-training. Recently, Yuan et al. (2025) proposed Native Sparse Attention (NSA), employing a dynamic hierarchical sparse strategy that integrates coarse-grained token compression with fine-grained token selection. NSA maintains global context awareness and local precision, enhancing training efficiency, inference speed, and model accuracy through hardware-aligned design. However, NSA has notable limitations: (1) it requires substantial training resources, complicating integration into existing well-trained models, and (2) its dynamic selection strategy prevents effective KV cache compression. Nevertheless, NSA's insights—particularly its block-wise selection strategy and the fusion of global context with fine-grained information—offer valuable directions for KV cache compression, which remain largely unexplored in current research.

We propose **H**ierarchical **B**lock- **W**ise **KV** cache compression (**HBW-KV**), which is training-free and universally compatible with existing LLM frameworks and inherits the advantages of both SnapKV and NSA. Building on SnapKV, we adapt NSA's sparse attention strategy to a training-free context, enabling plug-and-play integration with pre-trained models. Specifically, HBW-KV consists of two key components: (1) a block-wise selection mechanism, which empirically demonstrates higher precision than traditional token-level selection methods (as used by SnapKV and others), and (2) a hierarchical selection approach that addresses the typical concentration of selected positions at the beginning and end of prompts, thereby preserving more global context. Our HBW-KV captures comprehensive global information without introducing additional learnable modules, effectively bridging the gap between NSA-inspired sparse attention and practical KV cache compression.

Our contributions can be summarized as follows:

- We propose a block-wise selection method for KV cache compression that empirically outperforms token-level selection methods used in SnapKV and previous works.

- We identify the problem of imbalanced attention score distributions—often focused on prompt boundaries—and introduce a hierarchical selection strategy that better preserves global context without extra training.

- Extensive experiments across various LLMs and datasets demonstrate significant improvements in inference efficiency, memory usage, and accuracy over existing KV cache eviction methods, as shown in Figure 1. For instance, with 32K sequences, our HBW-KV method achieves a 16x compression ratio, reduces KV Cache by over 90%, accelerates decoding by $4\times$, and maintains over 99% of the original accuracy.

## 2 RELATED WORKS

**Sparse Attention Mechanisms.** Multi-Query Attention (MQA) Shazeer (2019) and Grouped-Query Attention (GQA) Ainslie et al. (2023) introduced a trade-off solution by dividing the query heads into multiple groups, while each group shares its own keys and values. MQA and GQA variants can optimize key-value (KV) management by intra-layer sharing to reduce redundancy. Recently, Native Sparse Attention (NSA) Yuan et al. (2025) employed a dynamic hierarchical sparse strategy, combining coarse-grained token compression with fine-grained token selection. As aforementioned, it has limitations: (1) It relies on heavy training and cannot be seamlessly integrated into existing models. (2) It dynamically selects important blocks based on incoming tokens, which prevents effective KV cache compression. In this paper, inspired by NSA, we propose a training-free NSA approach for KV cache compression. Our method is plug-and-play, offering significant accuracy improvements compared to existing KV cache eviction methods.

**KV Cache Eviction Methods.** Numerous studies have explored optimizing KV cache through selective retention. StreamingLLM Xiao et al. (2023) maintained only the initial tokens (attention sinks) and most recent tokens, but discard valuable intermediate information. H2O Zhang et al. (2023) used a scoring-based strategy that removes KVs during generation according to cumulative attention metrics. While effective for compressing KVs added during generation, this method fails to address prompt KV compression, which is essential for reducing both memory usage and computational demands. FastGen Ge et al. (2023) developed an Adaptive KV Compression system featuring a two-stage algorithm with four distinct compression policies. ScissorHands Liu et al. (2024) retained tokens with stable attention patterns across previous windows. However, both FastGen and ScissorHands

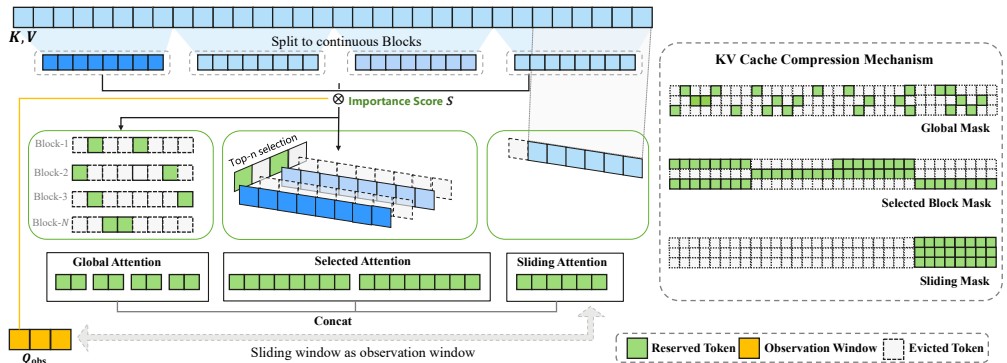

Figure 2: The workflow of our proposed HBW-KV. Left: KV cache is selected via three branches, including global selection for overall context, block-wise selection for important segments, and sliding attention for local context. Right: Visualization of attention patterns by each branch. Green areas indicate reserved regions while white ones represent skippable areas.

encounter limitations similar to H2O regarding prompt handling. In contrast, SnapKV Li et al. (2024b) focused on compressing KVs during the prefill stage by analyzing attention allocation, enabling efficient KV cache reduction for long prompts without sacrificing accuracy.

Our approach builds upon SnapKV, integrating the principles of NSA to further improve performance: (1) We propose a block-wise selection method, which empirically outperforms the token-level approaches used in SnapKV and prior works. (2) We observe that attention-based selection often leads to unevenly distributed indices, focusing on prompt boundaries and losing global context. To address this, we propose a hierarchical selection strategy—without adding extra learnable modules—to better preserve global information.

# 3 THE PROPOSED METHOD

In this section, we first briefly revisit the background and notations, and then introduce our proposed method. The overall framework of our method is shown in Figure 2.

## 3.1 BACKGROUND

The attention mechanism Vaswani et al. (2017), widely adopted in Transformer models, produces the attention output as a weighted sum of value matrix $\boldsymbol{V}$, where the weights are determined by the similarity between query matrix $\boldsymbol{Q}$ and key matrix $\boldsymbol{K}$. Formally, it can be expressed as:

$$\text{Attention}(\boldsymbol{Q}, \boldsymbol{K}, \boldsymbol{V}) = \text{softmax}\left(\frac{\boldsymbol{Q}\boldsymbol{K}^{\top}}{\sqrt{d}}\right)\boldsymbol{V}, \tag{1}$$

where $\boldsymbol{Q} \in \mathbb{R}^{L_q \times d}$, $\boldsymbol{K}, \boldsymbol{V} \in \mathbb{R}^{L_k \times d}$, in which $L_q$ and $L_k$ denote the lengths of query and key sequences, respectively, and $d$ indicates the hidden dimension.

To avoid redundant recomputations of $\boldsymbol{K}$ and $\boldsymbol{V}$ for previous tokens, these matrices are cached incrementally, which is termed KV cache. Mathematically, for a sequence of length $L_k$, the memory footprint of the KV cache grows as $O(2 \cdot L_k \cdot d)$, which scales linearly with sequence length. While KV cache reduces computational overhead from $O(L_k^2 \cdot d)$ to $O(L_k \cdot d)$ per step, it introduces significant memory pressure for long sequences.

## 3.2 HBW-KV FOR KV CACHE COMPRESSION

To enhance efficiency and reduce memory overhead, we propose a novel KV cache compression method, which contains three components, i.e., importance score computation, block-wise selection, and hierarchical block-wise selection.

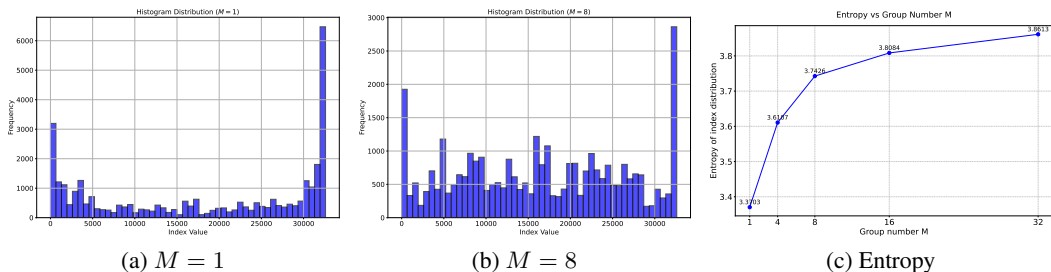

(a) $M = 1$           (b) $M = 8$           (c) Entropy

Figure 3: Effect of group number $M$ on index distribution and information coverage (illustrated with Mistral-7B-Instruct-v0.2). (a) and (b): Histograms for $M = 1$ and $M = 4$, showing improved uniformity as $M$ increases. (c): Entropy grows with $M$, reflecting enhanced global information capture through higher grouping diversity.

### 3.2.1 IMPORTANCE SCORE COMPUTATION

Following SnapKV, we divide the prompt sequence $L_k$ into two segments:

- **Observation Window** ($L_{\text{obs}}$), the last segment of the prompt, used to compute the importance scores.
- **Prefix Segment** ($L_{\text{prefix}}$), the initial portion of the prompt preceding the observation window. Overall, we have:

$$L_k = L_{\text{prefix}} + L_{\text{obs}}.\tag{2}$$

Given the query vectors from the observation segment $\boldsymbol{Q}_{\text{obs}} \in \mathbb{R}^{N \times L_{\text{obs}} \times d}$ and the key vectors from the prefix segment $\boldsymbol{K}_{\text{prefix}} \in \mathbb{R}^{N \times L_{\text{prefix}} \times d}$, where $N$ denotes the number of attention heads, the attention scores are computed as:

$$\boldsymbol{S} = \text{softmax}\left(\frac{\boldsymbol{Q}_{\text{obs}} \boldsymbol{K}_{\text{prefix}}^{\top}}{\sqrt{d}}\right) \in \mathbb{R}^{N \times L_{\text{obs}} \times L_{\text{prefix}}}.\tag{3}$$

Next, we aggregate the scores by summing along the observation window dimension $L_{\text{obs}}$ to obtain a tensor $\boldsymbol{S}_{\text{agg}}$:

$$\boldsymbol{S}_{\text{agg}} = \sum_{l=1}^{L_{\text{obs}}} \boldsymbol{S}[:,l,:],\tag{4}$$

where $\boldsymbol{S}_{\text{agg}} \in \mathbb{R}^{N \times L_{\text{prefix}}}$. Accordingly, we select $k$ reserved tokens based on tensor $\boldsymbol{S}_{\text{agg}}$ per head, where $k$ is referred to as the *KV cache capacity* and defined by $k = \lfloor p \cdot L_{\text{prefix}} \rfloor$, in which $p$ denotes the compression rate.

### 3.2.2 BLOCK-WISE SELECTION

We further divide the prefix segment into continuous blocks of size $T$. For each block $B_i \in \mathbb{R}^{N \times T}$, we compute a block-wise importance score by averaging the scores within the block:

$$\boldsymbol{S}_{\text{block}}(i) = \frac{1}{T} \sum_{j \in B_i} \boldsymbol{S}_{\text{agg}}[:,j].\tag{5}$$

We then select the blocks with the highest scores:

$$I = \text{Topk}\left(\boldsymbol{S}_{\text{block}}, \lfloor \frac{k}{T} \rfloor\right),\tag{6}$$

where operation $\text{Topk}$ selects indices $I$ of the top $\lfloor \frac{k}{T} \rfloor$ blocks in tensor $\boldsymbol{S}_{\text{block}}$ per head, with $k$ denoting the target KV cache capacity as previously defined. Consequently, the compressed KV cache is composed of the key and value vectors corresponding to the selected indices $I$, expressed as:

$$\boldsymbol{K}_{\text{compressed}}, \boldsymbol{V}_{\text{compressed}} = \boldsymbol{K}[I], \boldsymbol{V}[I].\tag{7}$$

The block-wise selection process is illustrated in Figure 2. Notice that our proposed block-wise selection in Eq. equation 6 has more feasibility; Eq. equation 6 actually degenerates into SnapKV when $T$ equals 1. Besides, NSA similarly employs block-wise selection; however, there are two key distinctions between our method and NSA:

1. Our approach directly calculates the importance score for each block based on the $\boldsymbol{K}$ vectors without training, whereas NSA computes scores using each block's summary vector via a trainable MLP.

2. We adopt a fixed $L_{\text{obs}}$ for selection, whereas NSA dynamically re-selects blocks per incoming token. Thus, our approach enables effective KV cache compression, while NSA is unable to perform KV cache eviction.

### 3.2.3 HIERARCHICAL BLOCK-WISE SELECTION

Previously, we introduced a KV cache compression strategy based on importance scores. It is worth noting that both our block-wise selection strategy and previous works Li et al. (2024b); Zhang et al. (2023) treat the entire KV cache as a whole when selecting. However, it is empirically observed that the selected indices $I$ often exhibit an imbalanced distribution, clustering notably at the head and tail positions, as shown in Figure 3a. Such imbalance may introduce bias due to discrepancies between the observation window and incoming tokens, leading to the potential loss of global information.

To more effectively preserve global information, we further evolve our method into Hierarchical Block-Wise KV cache compression (**HBW-KV**), as illustrated in Figure 2.

The core idea is to partition the prefix KV cache, $L_{\text{prefix}}$, into $M$ groups $G_1, G_2, \ldots, G_M$ and then apply block-wise selection independently within each group. Specifically, for a total capacity of $C$, we allocate an equal capacity of $C/M$ to each group to retain its most important tokens. This hierarchical design significantly enhances the coverage of global information. As illustrated in Figure 3b, increasing $M$ yields a more balanced token distribution. Correspondingly, Figure 3c shows that the entropy of this distribution grows with a larger $M$, further confirming that our approach achieves a more balanced representation of the input sequence.

Building upon this, we adopt a progressive selection procedure. The process begins with an initial round of block-wise selection where $M = 1$ to preserve the most critical global tokens. In subsequent rounds, we incrementally increase $M$ (e.g., to 8), thereby enriching the diversity of the selected tokens at different granularities. For the sliding window part, we directly employ the observation window for simplicity. The experimental results further demonstrate that our strategy outperforms alternative methods in capturing global context.

## 4 EXPERIMENTAL RESULTS

In this section, we conduct comparative and ablation experiments on benchmark datasets to verify the effectiveness of our proposed HBW-KV, which significantly reduces memory usage while preserving the most informative context for subsequent attention computations. All experiments are conducted using PyTorch on 4 Tesla V100 GPUs.

### 4.1 CONFIGURATIONS

Following the setting of SnapKV, we here conduct the long-text benchmark LongBench Bai et al. (2024) and Needle-in-a-Haystack test Kamradt (2023).

The proposed HBW-KV is built upon Huggingface's Transformers and PyTorch. We validate the performance of our mechanisms across various models that can handle extended contexts, including `LWM-Text-Chat-1M` (1M context), `LongChat-7b-v1.5-32k` and `Mistral-7B-Instruct-v0.2` (both 32k context). The first two are based on the LLaMA architecture Touvron et al. (2023), while the last one adopts the Mistral architecture Jiang et al. (2023).

Table 1: Performance comparison of our HBW-KV, SnapKV and H2O across various LLMs on LongBench. † denotes results from SnapKV. Best results in each cell are underlined.

| | | Single-Document QA | | | Multi-Document QA | | | Summarization | | | Few-shot Learning | | | Synthetic | | Code | | Overall |
|---|---|---|---|---|---|---|---|---|---|---|---|---|---|---|---|---|---|---|
| | LLMs * | NrtvQA | Qasper | MF-en | HotpotQA | 2Wiki | Musique | GovRep | QMSum | MultiNews | TREC | TriviaQA | SAMSum | PCount | PRe | Lcc | RB-P | Avg |
| LWMChat | All KV | 18.18 | 25.94 | 42.53 | 24.57 | 19.39 | 10.49 | 27.14 | 24.90 | 24.32 | 71.0 | 60.41 | 39.41 | 3.5 | 6.0 | 44.71 | 44.03 | 100% |
| | SnapKV: 1024 | 17.21 | 23.96 | 42.51 | 23.77 | 17.93 | 10.07 | 19.14 | 23.99 | 22.81 | 69.5 | 60.73 | 38.62 | 2.5 | 5.0 | 42.79 | 43.79 | 95.4% |
| | **Ours**: 1024 | 17.44 | 25.34 | 43.36 | 23.83 | 17.52 | 10.19 | 19.80 | 24.35 | 23.14 | 70.5 | 61.12 | 39.68 | 3.0 | 5.5 | 44.27 | 44.57 | 97.4% |
| | SnapKV: 2048 | 17.75 | 25.30 | 42.91 | 23.49 | 18.27 | 9.71 | 21.64 | 23.85 | 24.19 | 70.0 | 61.16 | 39.10 | 3.0 | 5.5 | 43.97 | 43.26 | 97.2% |
| | **Ours**: 2048 | 18.31 | 25.71 | 43.59 | 23.83 | 18.41 | 10.52 | 23.22 | 24.58 | 23.73 | 70.5 | 61.07 | 39.37 | 3.0 | 6.0 | 45.10 | 46.13 | 99.3% |
| | SnapKV: 4096 | 18.29 | 25.64 | 43.00 | 23.48 | 17.83 | 9.72 | 24.57 | 24.74 | 24.23 | 70.0 | 61.07 | 39.13 | 3.0 | 5.5 | 44.24 | 44.76 | 98.5% |
| | H2O: 4096† | 13.17 | 24.82 | 20.01 | 16.86 | 9.74 | 7.2 | 25.77 | 23.26 | 23.83 | 71.0 | 61.06 | 40.33 | 0.0 | 0.0 | 41.52 | 40.97 | 86.2% |
| | **Ours**: 4096 | 18.36 | 25.79 | 43.96 | 24.00 | 18.28 | 10.04 | 25.59 | 24.83 | 24.29 | 70.5 | 60.82 | 39.70 | 3.0 | 6.5 | 45.38 | 46.36 | 100.2% |
| LongChat | All KV | 22.61 | 34.32 | 46.10 | 38.78 | 27.20 | 16.03 | 30.88 | 22.57 | 26.39 | 66.5 | 83.99 | 40.61 | 0.5 | 30.5 | 54.82 | 58.92 | 100% |
| | SnapKV: 1024 | 21.21 | 30.24 | 40.41 | 36.31 | 25.75 | 14.01 | 22.95 | 21.48 | 25.01 | 61.5 | 77.75 | 38.19 | 0.5 | 30.0 | 52.81 | 57.41 | 92.5% |
| | **Ours**: 1024 | 21.66 | 32.27 | 45.49 | 38.22 | 26.16 | 14.42 | 24.50 | 21.56 | 25.74 | 65.0 | 82.82 | 38.72 | 0.5 | 30.5 | 52.74 | 57.93 | 96.3% |
| | SnapKV: 2048 | 21.13 | 33.65 | 41.54 | 36.36 | 26.41 | 14.67 | 25.33 | 22.26 | 26.14 | 65.0 | 77.05 | 39.28 | 0.5 | 30.0 | 54.93 | 57.94 | 95.3% |
| | **Ours**: 2048 | 22.23 | 34.21 | 44.77 | 38.94 | 26.58 | 14.82 | 27.12 | 21.94 | 26.28 | 65.5 | 82.20 | 39.97 | 0.5 | 31.0 | 55.07 | 58.04 | 98.1% |
| | SnapKV: 4096 | 21.67 | 35.08 | 43.96 | 37.07 | 27.06 | 14.37 | 28.22 | 22.83 | 26.16 | 65.5 | 80.13 | 40.02 | 0.5 | 30.5 | 54.82 | 58.69 | 97.7% |
| | H2O: 4096† | 19.31 | 28.30 | 37.75 | 30.51 | 23.06 | 11.76 | 27.55 | 21.37 | 26.49 | 66.0 | 75.80 | 39.92 | 0.0 | 25.5 | 53.56 | 55.53 | 90.3% |
| | **Ours**: 4096 | 22.70 | 35.20 | 45.75 | 38.84 | 26.98 | 15.20 | 28.43 | 22.77 | 26.34 | 66.5 | 82.75 | 41.08 | 0.5 | 31.0 | 54.28 | 58.92 | 99.4% |
| Mistral | All KV | 25.30 | 32.94 | 49.28 | 42.98 | 27.88 | 19.22 | 32.91 | 24.08 | 27.04 | 71.0 | 86.23 | 42.80 | 2.75 | 86.98 | 55.85 | 53.43 | 100% |
| | SnapKV: 512 | 23.49 | 27.69 | 48.35 | 40.44 | 25.90 | 17.00 | 23.08 | 23.55 | 24.05 | 65.5 | 85.99 | 41.08 | 2.74 | 87.13 | 54.2 | 51.48 | 94.3% |
| | **Ours**: 512 | 24.31 | 29.81 | 50.11 | 42.06 | 26.10 | 17.98 | 23.76 | 23.85 | 24.27 | 67.0 | 86.12 | 41.89 | 2.97 | 87.56 | 55.41 | 52.13 | 96.3% |
| | SnapKV: 1024 | 23.28 | 31.13 | 48.61 | 41.48 | 27.02 | 18.55 | 26.03 | 23.89 | 25.97 | 68.0 | 86.32 | 41.91 | 2.64 | 87.56 | 55.65 | 51.81 | 96.9% |
| | **Ours**: 1024 | 24.89 | 31.84 | 50.08 | 42.39 | 27.17 | 18.90 | 26.31 | 23.85 | 26.16 | 69.5 | 86.38 | 41.64 | 3.09 | 88.17 | 56.13 | 52.75 | 98.3% |
| | SnapKV: 2048 | 24.31 | 32.49 | 48.82 | 43.30 | 27.65 | 18.85 | 28.59 | 24.13 | 26.69 | 68.5 | 86.50 | 42.56 | 2.88 | 86.77 | 55.84 | 52.52 | 98.5% |
| | **Ours**: 2048 | 25.02 | 32.80 | 49.68 | 43.68 | 27.73 | 19.85 | 29.06 | 24.61 | 26.83 | 71.0 | 86.58 | 42.09 | 3.03 | 88.54 | 55.57 | 53.24 | 99.8% |
| | SnapKV: 4096† | 26.41 | 33.36 | 49.81 | 42.32 | 27.93 | 18.76 | 30.74 | 24.19 | 27.08 | 71.0 | 86.25 | 43.01 | 2.73 | 86.18 | 55.62 | 52.65 | 99.6% |
| | SnapKV: 4096 | 25.12 | 33.43 | 49.40 | 42.72 | 28.20 | 19.25 | 30.46 | 24.02 | 26.73 | 70.0 | 86.30 | 42.17 | 2.65 | 85.77 | 55.93 | 53.21 | 99.2% |
| | H2O: 4096† | 22.61 | 29.06 | 47.22 | 36.54 | 20.60 | 16.25 | 30.00 | 23.80 | 26.75 | 70.5 | 86.16 | 42.97 | 3.46 | 86.38 | 53.72 | 51.10 | 95.1% |
| | **Ours**: 4096 | 26.47 | 33.66 | 50.07 | 43.57 | 27.90 | 19.02 | 31.76 | 24.93 | 27.07 | 71.0 | 86.40 | 42.57 | 3.03 | 87.04 | 55.83 | 52.92 | 100.4% |

## 4.2 Experiments on Benchmark Datasets

As shown in Table 1, we evaluate HBW-KV on three models using LongBench, a multi-task benchmark for long context understanding, covering tasks like document QA, summarization, few-shot learning, synthetic tasks, and code completion. For each model, we compress the prompt KV cache to 1024, 2048, and 4096 tokens. The average input token length for these models is about 13k. Consequently, our method achieves an average compression rate of 92% with a cache size of 1024 tokens, and 68% with 4096 tokens. The last column of Table 1 reports the quantile of the average score across all datasets relative to the uncompressed baseline. Table 1 illustrates a negligible performance drop from models with our HBW-KV compared with original implementations for 16 different datasets. For instance, with the Longchat model, our HBW-KV achieves an accuracy of 96.3% using a KV cache size of 1024, compared to 92.5% for SnapKV. Notably, this even exceeds SnapKV's accuracy of 95.3% when using a KV cache size of 2048. These results confirm that HBW-KV effectively captures key information within long contexts and produces comprehensive, detail-rich summaries.

## 4.3 Generalizability on Needle-in-a-Haystack

In Sec. 4.2, we demonstrated that our method is compatible with diverse LLM architectures or weights. Herein, we assess its generalization capability on an additional dataset, the Needle-in-a-Haystack dataset. It challenges the model to accurately retrieve information from a specific sentence ("needle") concealed within an extensive document (the "haystack"), with the sentence placed at a random location. Typically, sentences that are inserted in the middle of prompts are harder to retrieve.

Figure 4 compares different compression methods on single-token retrieval with Mistral-7b-instruct-v0.2. With a KV cache size of 2048 (the second column), our HBW-KV achieves even higher average accuracy than full KV (86.94 vs. 85.32), whereas SnapKV suffers a 1-point loss. Moreover, our method's advantage becomes more pronounced with longer sequences. For example, at a length of 30K, our method achieves an accuracy of 81.43, while the baseline and SnapKV reach 80.65 and 74.54, respectively. Furthermore, at higher compression rates (with a KV cache size of 1024), the advantage of our method is further amplified.

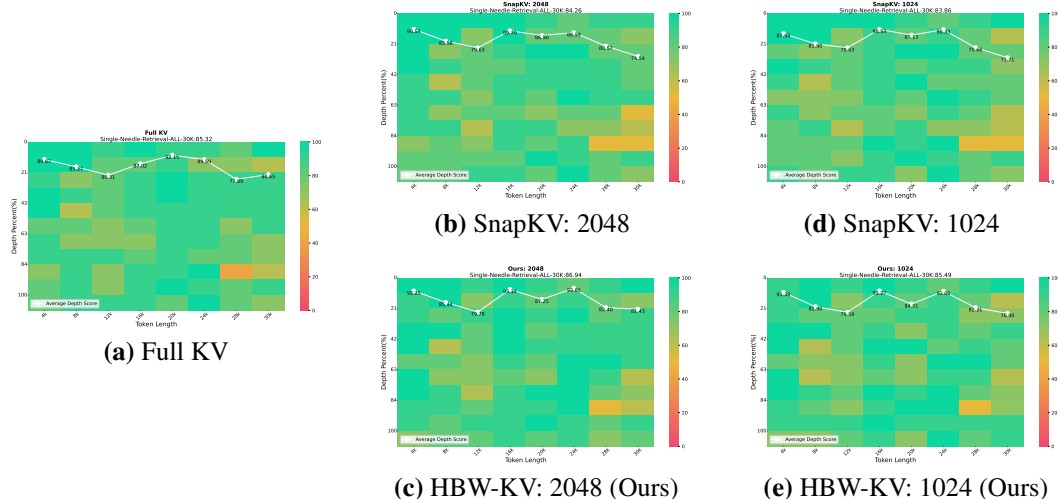

**(a)** Full KV

**(b)** SnapKV: 2048

**(d)** SnapKV: 1024

**(c)** HBW-KV: 2048 (Ours)

**(e)** HBW-KV: 1024 (Ours)

Figure 4: A comparison of different compression methods on the Needle-in-a-Haystack (single needle) dataset based on Mistral-7b-instruct-v0.2. Our HBW-KV achieves higher accuracy compared to SnapKV, and maintains lossless accuracy even when the KV cache size is 1024.

## 4.4 ABLATION STUDY

In this section, we conduct ablation studies to evaluate the impact of different components in our method, with Mistral-7B-Instruct-v0.2 as the target LLM. Firstly, we study the impact of block-size $T$ in block-wise selection. Then, we assess the contribution of incorporating global information. Finally, we investigate the combined effect of these two approaches, which together form our HBW-KV.

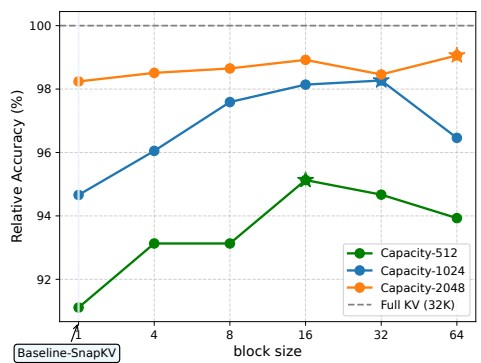

Figure 5: Relative accuracy on LongBench single-doc QA benchmark varies with block size and KV cache capacity. The x-axis represents block size, and the y-axis represents the relative accuracy compared to the full KV cache.

Figure 6: Performance comparison across datasets using various methods (KV cache capacity = 512) relative to the full KV baseline. The y-axis shows relative accuracies, and the x-axis represents different datasets on LongBench.

### 4.4.1 EFFECT OF BLOCK SIZES $T$

First we investigate the effect of our block strategy and notice that when block size $T$ equals 1, it actually degenerates into SnapKV. To provide a more intuitive representation, we report the relative accuracy compared to full KV in Figure 5. It is evident that the default block size of 1 is not the optimal choice (selecting only the most important token). Increasing the block size leads to a significant improvement in accuracy. For instance, with a KV cache capacity of 512, there is a 4% increase in accuracy (from 91.11% to 95.13%), which even surpasses the performance of SnapKV with a KV

Table 2: Effect of global information under KV cache capacity of 1024.

| Global Strategy | LongBench Single-Doc QA | | | | |
|---|---|---|---|---|---|
| | NrtvQA | Qasper | MF-en | MF-zh | **AVG** |
| Full KV | 25.30 | 32.94 | 49.28 | 58.47 | 100% |
| first | 21.92 | 24.04 | 34.75 | 32.57 | 68.2% |
| last | 4.33 | 16.37 | 14.08 | 15.85 | 30.5% |
| max | 18.29 | 17.89 | 21.32 | 23.54 | 48.8% |
| avg | 4.34 | 9.01 | 5.37 | 4.93 | 14.2% |
| $M = \{8\}$ | 22.48 | 27.4 | 45.2 | 47.53 | **85.9%** |
| $M = \{1\}$ (SnapKV) | 23.28 | 31.13 | 48.61 | 54.10 | 94.7% |
| $M = \{1, 8\}$ **(Ours)** | 24.32 | 31.76 | 48.42 | 55.32 | **96.3%** |

Table 3: Ablation study of our method using Mistral-7B-v0.2-Instruct as the target LLM.

| KV cache | | Block-wise Selection | Hierarchical Selection | LongBench Single-Doc QA | | | | | LongBench Few-Shot | | | | |
|---|---|---|---|---|---|---|---|---|---|---|---|---|---|
| Budget | Memory | | | NrtvQA | Qasper | MF-en | MF-zh | AVG | TREC | TriviaQA | SAMSum | lsht | AVG |
| Full 32K | 4096MB | × | × | 25.30 | 32.94 | 49.28 | 58.47 | 100% | 71.0 | 86.23 | 42.80 | 39.0 | 100% |
| 1024 | 128MB | × | × | 23.28 | 31.13 | 48.61 | 54.10 | 94.7% | 68.0 | 86.32 | 41.91 | 30.0 | 94.6% |
| | | × | ✓ | 24.32 | 31.76 | 48.42 | 55.32 | 95.7% | 69.0 | 86.16 | 41.67 | 32.0 | 95.7% |
| | | ✓ | × | 25.07 | 31.63 | 49.74 | 56.29 | 98.0% | 69.0 | 86.25 | 40.78 | 34.0 | 96.2% |
| | | ✓ | ✓ | 24.89 | 31.84 | 50.08 | 56.75 | 98.5% | 69.5 | 86.38 | 41.64 | 36.0 | **97.7%** |
| 512 | 64MB | × | × | 23.49 | 27.69 | 48.35 | 51.71 | 91.1% | 65.5 | 85.99 | 41.08 | 25.75 | 91.8% |
| | | × | ✓ | 23.61 | 28.68 | 49.27 | 52.68 | 92.9% | 66.0 | 85.99 | 42.27 | 27.5 | 92.8% |
| | | ✓ | × | 24.04 | 28.31 | 49.90 | 55.20 | 94.8% | 66.0 | 86.22 | 41.01 | 32.5 | 94.4% |
| | | ✓ | ✓ | **24.31** | **29.81** | **50.11** | **55.26** | **96.1%** | 67.0 | 86.12 | 41.89 | 34.5 | **96.0%** |

cache size of 1024. This indicates that the strategy of block-based selection effectively improves the compression accuracy under these settings.

This block strategy shows similar effects at different KV cache sizes. As the cache capacity increases, the performance gains gradually diminish, which is expected because SnapKV incurs only minimal losses with larger KV cache capacities.

Another interesting observation is that the optimal block size scales proportionally with the KV cache capacity, as shown by the star markers in Figure 5. Specifically, the optimal block sizes for capacities 512, 1024, and 2048 are 16, 32, and 64, respectively. Therefore, we empirically set the block size using the formula: Block Size = KV Cache Capacity / 32.

### 4.4.2 EFFECT OF GLOBAL INFORMATION

In this subsection, we investigate the impact of global information. In NSA, the KV cache is divided into multiple blocks, and each block generates a summary vector through an additional learnable MLP, which is regarded as global information. In this paper, our goal is to make it plug-and-play, so we need to explore training-free methods for retaining global information.

Previously we mentioned that the index distribution selected directly based on importance scores is imbalanced, which may lead to the loss of global information. This motivates us to explore the integration of global information. For fair comparison, we do not adopt block-wise selection ($T$=1).

There are some direct methods to generate global information, such as dividing $L_{\text{prefix}}$ into blocks, and produce one vector from each block as a representative by selecting the first element (first), selecting the last element (last), selecting the highest-scoring element (max), or taking the average within the block (avg). As illustrated in Table 2, all these methods incurs accuracy degradation.

To address this, we propose a hierarchical method for incorporating global information. When the group number $M$ is set to 1, it is equivalent to SnapKV. As shown in Figure 3, as $M$ increases, more global information can be retained. The hierarchical method with $M = 1$ and other values, such as $M = 8$, achieves the best performance in Table 2.

### 4.4.3 COMBINATION OF THE TWO

Here we conduct ablation experiments on our HBW-KV, which comprises two strategies: block-wise selection and hierarchical selection to incorporate global information. It is noteworthy that without these two strategies, HBW-KV essentially degenerates to SnapKV (the first row in each cell). The

column labeled "AVG" in Table 3 indicates the average relative quantile scores of the different datasets in comparison to the full KV baseline.

As shown in Table 3, both block-wise and hierarchical selection greatly improve accuracy, and combining them achieves the best result. For example, with a KV cache of 512, SnapKV achieves 91.1% and 91.8% on single-doc QA and few-shot learning datasets respectively. Adding block-wise selection raises accuracy to 94.8% and 94.4%, and adding hierarchical selection further boosts it to 96.1% and 96%, outperforming SnapKV even at a larger cache size of 1024.

In Figure 6, we present a more intuitive comparison across all datasets on LongBench using bar charts. Our HBW-KV consistently achieves the best performance, highlighting a similar trend across these datasets. Results in Table 3 and Figure 6 demonstrate the effectiveness of our proposed HBW-KV method. Remarkably, our approach does not incur additional memory overhead: the size of our KV cache remains identical to that of SnapKV. Furthermore, in the following section, we will also show that our method can achieve a significant acceleration ratio during the decoding phase.

## 4.5 LATENCY COMPARISON

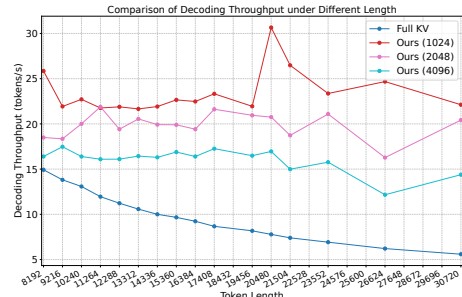

| Method | Speed (Tokens/s) | | KV Cache |
| --- | --- | --- | --- |
| | 16K | 32K | (MB) |
| Full KV | 9.68±0.02 | 5.74±0.01 | 4096 |
| Ours: 4096 | 16.70±1.17 | 17.08±3.20 | 512 |
| Ours: 2048 | 21.03±0.10 | 20.66±3.84 | 256 |
| Ours: 1024 | 22.08±1.52 | 22.48±2.96 | 128 |

Table 4: Comparison of decoding latency and KV cache memory usage at 16K and 32K context lengths.

Figure 7: Latency comparison under different prompt length using various methods.

Table 4 compares the decoding latency (tokens/s) and KV cache memory for various methods at different compression rates. We evaluated these on a Tesla V100 using Mistral-7B-Instruct-v0.2 with input sequences of 16K and 32K tokens. Results are averaged over five runs, with standard deviations reported. Figure 7 extends these performance metrics to a wider range of input sequence lengths.

As shown in Table 4 and Figure 7, since decoding is primarily memory-bound, inference speed with a full KV cache decreases nearly exponentially as input length increases. Our HBW-KV, however, effectively compresses the KV cache, yielding significant decoding speedups. For instance, with the KV cache compressed to 1024 tokens, our approach achieves a 4× speedup for 32K sequences.

Notably, results for SnapKV are not included. This is because our compression is applied post-prefill, and the KV cache size during decoding remains constant across all evaluated methods. Additionally, our method's contiguous selection of KV cache blocks theoretically enhances memory access efficiency, as discussed in NSA Yuan et al. (2025). While our current implementation does not optimize for this property, we reserve such optimizations for future work.

## 5 CONCLUSIONS

In this paper, we proposed the HBW-KV method for training-free KV cache compression in long-context large language models. The proposed HBW-KV contains the block-wise and hierarchical selection strategies that achieve superior precision over token-level methods while preserving global context without additional parameters. Extensive results demonstrated that the proposed method significantly improve the accuracy after compression while ensuring a high compression ratio and fast inference speed. In the future, it is attractive to further analyze the underlying principles behind the improvements led by our method and explore more effective compression patterns. In addition, it is also interesting to combine our approach with the distribution characteristics of different layers or attention heads for enhancing performance.

## ETHICS STATEMENT

The research presented in this paper focuses on the development of a compression algorithm. All experiments were conducted on publicly available datasets (LongBench and Needle-in-a-Haystack), which do not contain any personally identifiable or sensitive information. Our work does not involve human subjects, animal testing, or confidential data. To the best of our knowledge, we foresee no direct negative ethical implications or societal consequences resulting from this research.

## REPRODUCIBILITY STATEMENT

To ensure the reproducibility of our work, this paper fully discloses all necessary details to replicate the main experimental results. Section 4.1 provides a comprehensive description of the testing procedures, including dataset specifications, all hyperparameter values and other relevant configuration settings. While the source code is not publicly available at the time of submission, we believe the details provided are sufficient for independent reproduction of our findings. We commit to releasing the complete source code publicly upon acceptance of the paper.

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
