# OpenReview forum: "Training-Free Native Sparse Attention for KV Cache Compression"
_ICLR.cc/2026/Conference — Submitted to ICLR 2026_

### Official Review · Reviewer_dFjP · 2025-10-29

**Soundness:** 2
**Presentation:** 2
**Contribution:** 2
**Rating:** 2
**Confidence:** 4

**Summary:**

This paper introduces HBW-KV, a training-free method for KV cache compression in large language models. The approach adapts insights from Native Sparse Attention and features two main innovations: a block-wise selection mechanism that improves precision over token-level methods, and a hierarchical selection strategy to preserve global context without requiring extra training. The authors report significant results, including a 16x compression ratio on 32K sequences, over 90% reduction in KV cache size, a 4x decoding speedup, and accuracy maintained above 99%.

**Strengths:**

1. The method's training-free design facilitates straightforward integration with existing pre-trained models, bypassing the need for computationally expensive fine-tuning.
2. The empirical results demonstrate a strong balance of performance and accuracy, achieving significant decoding acceleration and KV cache compression while maintaining model fidelity above 99%.
3. The proposed block-wise and hierarchical selection strategies present a novel solution to the critical challenge of preserving global context, a known limitation of many existing cache compression methods.

**Weaknesses:**

* The paper does not quantify the computational overhead of the HBW-KV selection mechanism itself. This information is crucial for understanding the trade-offs involved and the net performance gain across different scenarios.
* The baseline included is not comprehensive. It's better to include the performance of the other existing methods, like PyramidKV, RocketKV, etc., in the comparison to better understand the benefits of HBW-KV.
* The method may have difficulty in handling multi-round conversation tasks. It lacks discussion on how the method performs on multi-turn conversation benchmarks.

**Questions:**

1. Could you provide a detailed analysis of the computational overhead from the block-wise and hierarchical selection strategies? A cost breakdown relative to the main attention computation would be particularly insightful for evaluating the net performance gains.
2. Could you report the peak memory usage during inference instead of just the KV cache memory cost?

---

### Official Review · Reviewer_e5L3 · 2025-10-30

**Soundness:** 3
**Presentation:** 3
**Contribution:** 3
**Rating:** 8
**Confidence:** 3

**Summary:**

This paper presents a practical and effective solution for reducing the memory and computational cost of large language model inference through a hierarchical block-wise KV cache compression technique. The method adaptively preserves both local and global attention information while removing redundancy, achieving over 90% KV cache reduction, 16× compression, and up to 4× faster decoding all without retraining or fine-tuning. Its design is training-free, hardware-friendly, and easy to integrate into existing architectures, making it a valuable contribution to efficient long-context processing. While the work is empirically strong, it could benefit from deeper theoretical analysis, robustness evaluation across diverse architectures, and variance reporting. Overall, HBW-KV represents a well-executed engineering innovation with high practical relevance for scaling LLMs efficiently.

**Strengths:**

Combining block-wise selection and a hierarchical strategy to preserve both local and global attention information serves as the bridge to SnapKV (NeurIPS 2024) and Native Sparse Attention (NSA) (2025). It averages scores over contiguous chunks, ranks chunks, and keeps whole high-value blocks. Empirically, in their Table 1 / Table 3, this gives higher accuracy than SnapKV at the same KV budget, sometimes even beating SnapKV, which allows a larger cache. The approach is training-free and plug-and-play, making it easy to integrate into existing pre-trained models without additional tuning a major strength compared to prior sparse or learned compression methods.

**Weaknesses:**

The weakness is that the paper focuses mainly on inference efficiency, with limited exploration of theoretical bounds, cross-architecture generalization, and prefill-phase optimization. Moreover, while the experiments are extensive, the paper could be improved by including variance analyses or testing on more diverse long-context benchmarks to confirm robustness.

They should report head-to-head wall-clock latency vs SnapKV on the same GPU and same model not just % compression and tokens/sec in isolation

**Questions:**

1. How does the method handle cases where the attention pattern is very spread out? Does it still keep enough important information when compressing the KV cache?
2. Can the authors explain how well the method works on larger or different models, like Llama-70B or multimodal models? Would any adjustments be needed?
3. Can you also add the head-to-head wall clock latency vs Snap KV on the same GPU and model(not just % compression and tokens/sec in isolation). It's important to quantify as paper reflects the KV Cache size during the decode ends up equal post prefill.
While  decoding throughput scaling and ~4× speedup at 32K context on V100 for Mistral-7B, which is credible, but adding the comparison to SnapKV in that latency plot will further strengthen the results

---

### Official Review · Reviewer_XTnv · 2025-10-31

**Soundness:** 3
**Presentation:** 2
**Contribution:** 2
**Rating:** 4
**Confidence:** 3

**Summary:**

This paper addresses the significant memory and latency bottlenecks caused by the Key-Value (KV) cache in long-context Large Language Model (LLM) inference. The authors propose HBW-KV (Hierarchical Block-Wise KV cache compression), a novel training-free method designed to compress the prompt KV cache, making it "plug-and-play" for existing pre-trained models.

The method builds upon prior work like SnapKV but introduces two key innovations inspired by Native Sparse Attention (NSA):

1. Block-wise Selection: Instead of selecting important tokens individually (token-level), HBW-KV groups tokens into contiguous blocks, calculates an average importance score for each block, and retains the top-scoring blocks。

2. Hierarchical Selection: The paper identifies that standard attention-score-based selection causes retained tokens to cluster at the boundaries of the prompt, neglecting global context. To solve this, HBW-KV partitions the context into $M$ distinct groups and applies block-wise selection independently within each group.

**Strengths:**

The paper tackles a highly significant and practical problem. The "memory-bound" nature of long-context decoding is a primary barrier to the widespread deployment of these models. A solution that provides a 4x decoding speedup while maintaining (or even improving ) accuracy is a very valuable contribution to the community.

**Weaknesses:**

See Questions below.

**Questions:**

1. Missing Latency Baseline: Could you please clarify why SnapKV was omitted from the latency benchmarks in Table 4 and Figure 7? A key claim is that HBW-KV is superior to SnapKV. A direct comparison of (HBW-KV @ 1024) vs. (SnapKV @ 1024) on both accuracy and latency would be crucial. Alternatively, comparing the latency of (HBW-KV @ 1024) to (SnapKV @ 2048) would also be compelling, as Table 1 suggests they have similar accuracy

2.Hyperparameter Sensitivity: The hyperparameters $T=\text{Capacity}/32$ and $M=\{1, 8\}$ are derived from your experiments. How sensitive is the model's accuracy to these settings? If a user has a new model or a different capacity target, do you recommend a full ablation sweep, or are these heuristics robust?

3. Do you have any preliminary projections on how much additional speedup a custom kernel that exploits this block-wise memory access pattern could provide, beyond the 4x speedup already achieved from token reduction alone?

4. The idea is incremental. Could you demonstrate your novelty?

---

### Official Review · Reviewer_M83D · 2025-11-01

**Soundness:** 2
**Presentation:** 1
**Contribution:** 1
**Rating:** 2
**Confidence:** 4

**Summary:**

The paper proposes HBW-KV, a training-free method for compressing the KV cache during the prefill stage. The method extends SnapKV, utilizing an observation window to score tokens. It introduces two main modifications inspired by Native Sparse Attention (NSA): 1) Block-wise selection, which selects continuous blocks of tokens rather than individual tokens, and 2) Hierarchical selection, which divides the context into groups and selects important blocks from each, aiming to preserve a broader global context.

**Strengths:**

The approach explores structured sparsity (block-wise selection) to potentially improve the distribution of retained tokens compared to the base SnapKV method.

**Weaknesses:**

**Q1**  The novelty of the proposed method is incremental. HBW-KV is essentially SnapKV augmented with two straightforward heuristics: block-wise grouping and hierarchical grouping. While inspired by NSA. NSA relies on learned summary vectors and a trainable MLP for block scoring. HBW-KV replaces these learned components with simple averaging (Eq. 5) and heuristic grouping. The method is more accurately described as heuristically structured SnapKV rather than a significant advancement in "Native Sparse Attention."

**Q2** The paper's evaluation is weakened by relying almost exclusively on SnapKV as the primary baseline. SnapKV is significantly outdated. Numerous highly relevant, superior, and recent SOTA works from 2025 are completely omitted from the comparisons. The authors should compare against leading training-free methods such as:
KVzip (NeurIPS 2025), FlexPrefill (ICLR25), MInference (Neurips2024)
and KVPress (https://github.com/NVIDIA/kvpress?utm_source=chatgpt.com)

**Q3** The paper evaluates on Mistral-7B a model utilizing Grouped-Query Attention (GQA). However, the methodology is described using standard Multi-Head Attention (MHA) notation, referring to N attention heads. The authors never explain how HBW-KV handles this critical architectural feature. Are the attention scores from all sharing Q heads aggregated to create one mask for the shared K/V head?

**Q4** The paper proposes a formula, "Block Size = KV Cache Capacity / 32" , derived by observing optimal sizes for only three specific capacities (512, 1024, 2048). This is a severe over-generalization from sparse data and cannot be considered a robustly derived heuristic.

**Q5** It is unclear how the method performs in multi-turn dialogue settings (e.g., using SCBench or multi-tuen in NIAH), where the KV cache grows dynamically and coherence across turns is critical.

**Q6** The paper does not specify the handling of tokens generated after the initial prompt compression. Are they all retained, or is a sliding window applied?

**Q7** The experiments focus on medium-to-long contexts (average 13k in Table 1, up to 32k in Table 4). The effectiveness of the compression on very long contexts (128K+), where bottlenecks are severe, and on shorter contexts (e.g., 128-256 tokens), is not evaluated.

**Q8** The hierarchical and block-wise selection strategies involve additional computation (averaging, sorting across blocks and groups) compared to SnapKV.  Did the author incorporate all these overheads?

**Q9** While the authors mention that contiguous blocks theoretically enhance memory access efficiency. They admit their implementation is not optimized for this. Therefore, the reported speedups are merely a function of having a smaller cache size, identical to what SnapKV would achieve with the same compression ratio, and not an inherent advantage of the HBW-KV structure.

**Q10** Typos
Redundancy: "Eq. equation 6".

**Questions:**

I already mentioned them in the Weaknesses section. The authors need to address it.

---

### Meta-Review · Area_Chair_irsd · 2025-12-07

**Summary:**

This paper proposes HBW-KV, a training-free KV cache compression method for long-context LLM inference. Building on SnapKV and inspired by Native Sparse Attention (NSA), the method introduces (1) block-wise selection of contiguous tokens and (2) hierarchical grouping of the context to better preserve global information. Experiments on Mistral-7B show substantial KV cache reduction (up to 16X), claimed decoding speedups (up to 4X), and largely preserved accuracy on long-context benchmarks.

Across four reviews, assessments are mixed. One reviewer is strongly positive and views the work as a practical, well-executed engineering contribution with high relevance, while the other three reviewers raise substantial concerns about novelty, experimental completeness, and methodological clarity. Two reviewers recommend rejection, one is marginally below the acceptance threshold, and one recommends acceptance. Overall, the consensus leans negative, primarily due to unresolved questions about novelty relative to prior work, missing and outdated baselines, and insufficient clarity about several key technical and experimental aspects.

**Reviewer Concerns:**

Unresolved:
* The method is viewed as an incremental extension of SnapKV using heuristic block-wise and hierarchical grouping.
* The paper does not explain how HBW-KV handles Grouped-Query Attention (GQA), despite evaluating on Mistral-7B.
* The block-size rule is based on very limited empirical evidence and lacks robustness.
* There is no evaluation on multi-turn dialogue, very short contexts, or extremely long contexts (128K+), and no direct wall-clock latency comparison with SnapKV.
* Additional computation from block-wise and hierarchical selection is not measured, making net speedups unclear.

**Reviewer Scores:**

* Reviewer M83D (2): unchanged
* Reviewer dFjP (2): unchanged
* Reviewer XTnv (4): unchanged
* Reviewer e5L3 (8): unchanged

---

### Decision · Program_Chairs · 2026-01-26

Reject